# Deep Learning-Based Segmentation of Head and Neck Organs-at-Risk with Clinical Partially Labeled Data

**DOI:** 10.3390/e24111661

**Published:** 2022-11-15

**Authors:** Lucía Cubero, Joël Castelli, Antoine Simon, Renaud de Crevoisier, Oscar Acosta, Javier Pascau

**Affiliations:** 1Departamento de Bioingeniería, Universidad Carlos III de Madrid, 28911 Madrid, Spain; 2Université Rennes, CLCC Eugène Marquis, Inserm, LTSI-UMR 1099, F-35000 Rennes, France; 3Instituto de Investigación Sanitaria Gregorio Marañón, 28007 Madrid, Spain

**Keywords:** DL, automated segmentation, head and neck radiotherapy, organs-at-risk, partially labeled, longitudinal data

## Abstract

Radiotherapy is one of the main treatments for localized head and neck (HN) cancer. To design a personalized treatment with reduced radio-induced toxicity, accurate delineation of organs at risk (OAR) is a crucial step. Manual delineation is time- and labor-consuming, as well as observer-dependent. Deep learning (DL) based segmentation has proven to overcome some of these limitations, but requires large databases of homogeneously contoured image sets for robust training. However, these are not easily obtained from the standard clinical protocols as the OARs delineated may vary depending on the patient’s tumor site and specific treatment plan. This results in incomplete or partially labeled data. This paper presents a solution to train a robust DL-based automated segmentation tool exploiting a clinical partially labeled dataset. We propose a two-step workflow for OAR segmentation: first, we developed longitudinal OAR-specific 3D segmentation models for pseudo-contour generation, completing the missing contours for some patients; with all OAR available, we trained a multi-class 3D convolutional neural network (nnU-Net) for final OAR segmentation. Results obtained in 44 independent datasets showed superior performance of the proposed methodology for the segmentation of fifteen OARs, with an average Dice score coefficient and surface Dice similarity coefficient of 80.59% and 88.74%. We demonstrated that the model can be straightforwardly integrated into the clinical workflow for standard and adaptive radiotherapy.

## 1. Introduction

Head and neck (HN) cancer is the seventh most common cancer worldwide. In 2020, 931,931 new patients were diagnosed with HN cancer, increasing the prevalence of the disease to 2,411,687 patients and accounting for 467,125 deaths [1]. Tumors can appear in various subsites, including the lip, oral cavity, salivary glands, nasopharynx, oropharynx, hypopharynx, larynx, nasal and paranasal cavity, and ear [2]. 

Radiotherapy (RT) is currently one of the most common treatment strategies. It consists of applying high-energy radiation from x-rays, gamma rays, or protons to the tumor mass and high-risk areas to damage the DNA of cancerous cells and impede their cellular division. Despite its significant advantages, the main limitation of RT is that the healthy tissues surrounding the tumor (organs at risk (OAR)) also suffer partial irradiation, which may lead to diverse complications and toxicities. Thus, one of the most critical steps during RT treatment planning is accurately contouring the target volume and OARs. This allows estimating the dose these structures will receive and developing personalized strategies to mitigate the radio-induced toxicities [3], which is critical in HN cancer due to the high amount of OARs present in the region.

### 1.1. Contouring in HN Radiotherapy

Contouring is often performed manually by trained radiation oncologists [4]. This task, also referred to as segmentation or delineation, is highly time-consuming and presents a subjective component [5,6,7,8]. An expert can spend over four hours on a single HN case [3,9], which may delay treatment increasing risks of local recurrence and mortality. Furthermore, this process is thoroughly subjected to inter- and intra-practitioner variabilities [6,9,10] driven by diverse factors such as experience, availability, quality, and interpretation of diagnostic imaging [4]. The limited soft tissue contrast of computed tomography (CT) images is also a substantial problem in HN delineation [10,11], as many OARs have similar densities to fat, muscle, or other surrounding tissues. Segmenting these OARs, such as the parotid or submandibular glands, is particularly challenging [8]. Moreover, various studies have associated contouring variation with decreased plan quality, worse disease control, increased toxicity, and inferior survival rates [4,9]. The limitations of manual segmentation also represent a barrier to adaptive radiotherapy [3]. In this context, automated computer-performed segmentation (auto-segmentation) has been explored to address all of these challenges. The following section presents the most important contributions to solving this problem in HN radiotherapy.

### 1.2. Literature Review

Until the past years, most of the automatic contouring tools in clinical use were atlas-based [9,12]. However, the anatomical variability between patients and the low grayscale contrast in certain HN areas yield errors in the atlas registration and, therefore, in the automatic segmentations [13]. In recent years, DL-based algorithms have proven to overcome the limitations of manual and atlas-based contouring, boosting performance [12,14,15], yielding more robust OAR segmentations and, therefore, safer and more efficient treatments [16,17,18]. Regardless, even if DL-based OAR auto-segmentation becomes the new standard, an expert must always examine the contours to account for specificities in the patient’s anatomy and tumor morphology. In this scenario, the clinicians would only need to review the delineations instead of performing a full segmentation, reducing the time spent by more than 90% [16]. Several DL methods have been proposed focused on HN contouring [19], predominantly based on convolutional neural networks (CNNs). Many of them are based on 3D U-Net [20] and achieve good performance [3,10,13,21,22]. Others are based on on a two [23,24,25,26] or multi-step [27] workflow. The number of structures studied ranges from 4 to 28 OARs. 

Several factors still limit the clinical implementation of DL-based auto-segmentation [6,9,28], including a lack of standardization of contouring protocols [10,29], trust among the users, and limited availability of large, labeled databases. Ideally, to train a robust DL segmentation network, an extensive database of patient CT images would need to be labeled, reviewed, and curated by several experts following the same delineation guidelines [7,30]. A second labeled database would allow for external validation. Nonetheless, this ideal scenario is highly unfeasible for most medical centers, as labeling large databases is incompatible with the daily clinical workload. On the contrary, clinical centers do have many partially labeled and unlabeled datasets [7,9]. In many cases, longitudinally collected data from the treatment is also available. These data can also be exploited with DL and, furthermore, used to optimize individual treatment for patients in an adaptive RT framework [5]. Curating this data may be necessary to ensure homogeneous learning [8,13,30,31].

Aiming to exploit the available data in a medical center, Liu et al. [11] proposed a multiview contrastive representative learning framework. They generated 2D CT slices as the input to a self-supervised 2D CNN outputting three 3D segmentations and using a contrastive loss to select the correlated predictions. They trained their network with more than 3D 200 CTs from unique patients, but only 56 were labeled. They used the labeled data for initial training and then fine-tuned the network using the unlabeled images, evaluating their results in the labeled dataset, achieving a mean Dice score coefficient (DSC) of 0.86 over 24 OARs. Although they made direct comparisons with other state-of-the-art methods, they lacked an external validation on an independent test set. Chi et al. [7] also proposed a solution to train CNNs with limited labels based on a weakly supervised algorithm with a pseudo-contouring generation technique. They generated pseudo-labels with a Demons-based free-form deformable image registration algorithm to transfer contours from labeled data to new images and then trained a recursive ensemble organ segmentation model. They segmented 5 OARs training their framework with only 29 labeled images, achieving DSCs between 0.61 and 0.88. AnatomyNet [32] was also trained with inconsistent data annotations with missing ground truth for some anatomical structures. As a solution, they employed a weighted loss function updating and balancing the weights according to the number of missing annotations, segmenting 9 OARs with a mean DSC of 0.79.

In this context, we propose a two-step solution to exploit the available partially labeled data in a medical center and train a robust DL-based auto-contouring framework for HN OAR segmentation. The method takes advantage of the longitudinal character of our database which allows to: (i) generate pseudo-contours for the missing labels; and (ii) improve segmentation accuracy by enriching the framework when incorporating a baseline image from each patient. Our solution performs robust segmentation of fifteen HN OARs and could be easily implemented for adaptive RT.

## 2. Materials and Methods

### 2.1. Database

Data were selected retrospectively from the ARTIX study (Adaptive Radiotherapy to Decrease Xerostomia in Oropharynx Carcinoma) [33]. All the patients were adults (>18 and <75 years of age) with locally advanced non-metastatic carcinoma of the oropharynx limited to T3 and T4 and N2-N3 treated with arc-IMRT (Intensity-Modulated Radiation Therapy). Data comprised HN CT scans from 48 patients treated with RT in the CLCC Eugène Marquis between 2013 and 2018. For each patient, all contours were delineated by a single expert radiation oncologist amongst a team of ten who participated in the whole study. The patients underwent adaptive RT with weekly replanning, with between two and six CT scans acquired and delineated during the treatment course.

Two hundred and sixty-nine CT scans were included in our study. The standard CT in-plane pixel spacing was 1.131 mm × 1.131 mm, whereas slice thickness varied between 2 mm and 3 mm. The clinical data comprised inconsistent data annotations with missing ground truth contours for some anatomical structures. In other words, a different number of OAR (between 5 and 30) were segmented in each CT image. In some cases, even for the same patient, particular OARs were delineated in some CT scans but not others. We selected the fifteen HN OARs with the highest availability throughout the cohort: brain, brainstem, right/left (R/L) inner ear, R/L parotid glands, R/L temporomandibular joint, mandible, R/L submandibular glands, lips, larynx, esophagus, and spinal canal. OAR contouring frequency is depicted in Table 1. From the 24 patients for which the described fifteen OARs were fully contoured in all CT scans, we randomly selected a subset of 8 patients. This yielded 44 CT images that were excluded from all training steps and saved as the independent test group.

### 2.2. Proposed Framework

We exploited the longitudinal information in our dataset to solve the partial annotations problem with the following two-step workflow (Figure 1). After preprocessing, we firstly trained a single-class OAR-specific 3D U-Net-based network for each organ. These trained models were used to generate pseudo-contours for the missing OAR delineations, obtaining multi-class masks for every CT in the training set. Secondly, a nnU-Net [34] was trained with the constructed multi-class masks. This final model exploits the anatomical spatial interdependence among the OARs, yielding the final delineations. We evaluated the workflow in the independent test group. The experiments were performed on a CUDA-enabled Nvidia Quadro RTX 8000 GPU with 48 GB of memory. DL-based models were implemented in PyTorch 1.11.

#### 2.2.1. Preprocessing

CT scans were resampled to a standard voxel space of 1 mm × 1 mm × 1 mm. Then, the maximum patient body size was calculated with a windowing technique that allowed removing the medical bed from the image and locating body edges. The maximum body size in each direction was computed and used to crop all CTs to a fixed size of 224 × 224 × 224 voxels, ensuring that all OARs were included in the resulting volume. The ground truth delineations of four OARs were curated under the supervision of a clinical expert according to the Brouwer Atlas [35], both in the training and test sets. The spinal canal and esophagus contours underwent curation to ensure that their cranial and caudal limits were consistent. Temporomandibular joints and inner ears required more extensive refinement to correct particular contours that deviated from the segmentation protocol. Moreover, the right and left contours of the symmetric organs (inner ears, temporomandibular joints, parotid, and submandibular glands) were joined into one single mask and considered as a single OAR. An ablation study conducted by our group (in artificial intelligence an ablation study allows to investigate the performance of an overall system when removing or modifying certain components, demonstrating an increased segmentation accuracy when dealing with symmetric OARs as a single anatomical structure [31,36]. This yielded eleven separate OARs to train the framework.

#### 2.2.2. OAR-Specific Models

A single-class network was trained to segment each OAR independently (OAR-specific models), all with the same architecture and training parameters: 3D Residual U-Net [37] with six levels; two residual units concatenated in each downsampling and upsampling layer; encoder with strided convolutions of size two and decoder image upsampling with strided transpose convolutions; and parametric rectifying linear units [38] and instance normalization to boost segmentation accuracy [37]. 

For each OAR-specific network, training, validation, and test sets were randomly built following an optimal division of 64%, 16%, and 20%, respectively. As each CT contained a different number of ground truth segmentations, the number of images to train each OAR-specific network also differed (Table 1). 

Extensive data augmentation, performed within MONAI [39] (a software framework for medical image analysis with deep learning that includes specific functions for medical data input, preprocessing, or augmentation), was applied to the CT and segmentation map at each epoch of the training to increase data variability and heterogeneity, including random 90° rotations, flips, and intensity shifts with a probability of 50%. Four random crops of 128 × 128 × 128 voxels, including the ground truth label, were performed for each input image at every epoch. The batch size was set to two, yielding eight samples per iteration. During validation, no data augmentation was applied. Inference was calculated with a sliding window of size 128 × 128 × 128 with a 25% overlap between scans. 

The loss function was a combination of Dice Coefficient and Cross-Entropy [14]. Models were trained and regularized with AdamW, a stochastic optimizer that decouples the L2 regularization weight decay in the Adam optimizer from the gradient update [40]. Each network was trained for 350 epochs with an initial learning rate of 10^−4^ and a weight decay of 10^−5^, completed in an average of 3 h and 24 min. The trained OAR-specific models were evaluated on their own test set and on the independent test group by computing the Dice score coefficient (DSC) and average surface distance (ASD) between the predictions and the ground truth masks. The DSC is defined as the relative spatial overlap between two binary volumes [41]; whereas the ASD is the average of all the Euclidean distances between the boundary voxels of two volumes [42].

An ablation study was performed to analyze the impact on the accuracy of having fewer CT images for training. As depicted in Table 1, the radiologists contoured the brainstem, spinal canal, mandible, and parotid glands in almost all CTs. On the contrary, the brain was segmented in only 172 CTs out of 225 samples. To analyze if this could have downgraded the performance of some OAR-specific models, we randomly removed 10, 20, 30, 40, and 50 samples from the brainstem, spinal cord, mandible, and parotid glands’ training sets. We trained a new OAR-specific model for each of these scenarios (reduced-data models) and compared their accuracy on the independent test set to the original models. The number of training images was different for each reduced-data model, and can be calculated by subtracting 10, 20, … to the values indicated in Table 1 (third column). 

#### 2.2.3. Pseudo-Contour Generation

We used the trained OAR-specific models to generate pseudo-contours for the CT images with missing labels to have a fully segmented training image set. We hypothesized that all patients would be contoured with enough precision by our networks, especially those for which a certain OAR was annotated on at least one CT image. After visual inspection, no meaningful errors or outliers were discovered among the generated pseudo-contours. The average time for pseudo-contour generation was 8 s per contour on GPU. A multi-class mask was created for each CT image, ensembling the available ground truth segmentations with the generated pseudo-contours replacing the missing labels, yielding labelmaps with eleven classes.

#### 2.2.4. Multi-Class Network

nnU-Net [34] is a deep learning-based segmentation framework. It is a self-configuring method that optimizes all the steps involved in segmentation for a new given task: preprocessing, network architecture, training parameters, and post-processing. We trained a 3D full-resolution U-Net within this framework with 5-fold cross-validation for 800 epochs per fold, not modifying any other default parameter in nnU-Net implementation. As input, we used the CTs and ensembled multi-class contours, and trained the network in a multi-class fashion to segment all OARs simultaneously. This allowed the model to learn and exploit the anatomical spatial relationships between the individual OARs, which was expected to be highly beneficial for segmentation robustness. The model was trained with a combination of Dice and cross-entropy losses and the Adam optimizer with an initial learning rate of 10^−4^. Each fold was completed in 23 h and 14 min on average.

The inference was implemented using an overlapping sliding window. The Dice score coefficient (DSC), surface Dice similarity coefficient (sDSC), and average surface distance (ASD) were computed to evaluate the performance of the trained model. The sDSC evaluates the overlap between two surfaces at a specified tolerance [3], so a perfect sDSC would imply that approximately 95% of the surface is correctly outlined whereas 5% must be refined. We implemented the sDSC using pre-computed tolerances presented in [3]. All the experiments were evaluated before and after data curation in the test set.

We conducted two experimental studies exploiting our longitudinal data. First, we evaluated the feasibility of adapting the model to new data for adaptive RT, following a similar approach to [43]. With this aim, we added the first acquired CT of each test patient to the training set and performed transfer learning with the trained nnU-Net for 100 new epochs. This model (self-supervised model) was again evaluated in the independent test set. Secondly, the accuracy of the predicted contours was analyzed per patient with the semi-supervised and self-supervised models.

## 3. Results

### 3.1. OAR-Specific Models

Table 2 depicts the evaluation of each OAR-specific model in its own test and independent test sets before and after data curation. The results were highly accurate in the OAR-specific test sets, with the DSC above 81% for all OARs except the lips and submandibular glands. On the independent test set, the DSC was still higher than 81% for the brain, mandible, spinal canal, and brainstem. On the contrary, the larynx, lips, and submandibular glands presented the most substantial deviations (66.73 ± 9.31%, 45.17 ± 18.18%, and 55.40 ± 21.14%, respectively). As expected, the four refined OARs underwent a substantial increase in performance after data curation, especially significant for the temporomandibular joints and inner ears (from 67.26% to 83.34%, and 55.49% to 83.76%, respectively). These results were consistent with the ASD evaluation.

Figure 2 shows the results from the ablation study with the reduced-data models: a comparison between the DSC obtained with the original OAR-specific models of the brainstem, spinal canal, mandible, and parotid glands, and the reduced-data ones. For all OARs, removing up to 40 training images did not affect the models’ performance in the independent test group. After extracting 50 images, a slight decrease in the DSC could be observed for the brainstem and the parotid glands, concurrently with a higher presence of outliers.

### 3.2. Multi-Class Network

Table 3 (a) shows the evaluation of our trained nnU-Net on the independent test group. The average DSC was 73.74 ± 18.61% for the uncurated data, and 80.59 ± 16.41% after curation of the ground truth contours. The four curated OARs experienced a substantial improvement after data homogenization (Figure 3), especially the temporomandibular joint and inner ear contours (the DSC improved from 67.92% to 89.73%, and from 54.74% to 86.03%, respectively). After data curation, all OARs presented a mean DSC above 81% except for the larynx, lips, and submandibular glands. The submandibular glands’ delineations comprised the more noticeable outliers, with a standard deviation in the DSC of 25.61%. Other significant outliers disappeared after data refinement, as in the case of the esophagus and the temporomandibular joints (Figure 3).

The sDSC was higher than the DSC for all OARs except the brain, for which the average sDSC and DSC were 96.66% and 97.99%, respectively. All OARs achieved a mean sDSC above 90%, except for the larynx, lips, and submandibular glands. The sDSC was particularly higher than the DSC for the parotid glands, temporomandibular joints, and inner ears, with mean values of 96.29%, 99.27%, and 96.98% after data curation. 

The predictions of the four refined OARs experienced a large reduction in their mean ASDs compared to the ground truth contours after curation (Figure 3). For the spinal canal, ASD decreased from 0.864 to 0.696 mm, and from 2.694 to 0.569 mm in the esophagus. The submandibular glands also showed significant outliers in the ASD, in agreement with the obtained DSC and sDSC. 

The numerical results of the self-supervised study conducted by retraining our model after including the first acquired CT image from the eight test patients are gathered in Table 3 (b). Compared to the semi-supervised model, the average DSC and sDSC increased for all OARs (84.26% and 93.01%, respectively). This improvement was especially noticeable in the larynx, lips, and submandibular glands, which were the OARs that obtained the worst performance with the original semi-supervised model. The larynx sDSC reached 85.67%, whereas the lips and submandibular glands increased to 78.83% and 83.79%, respectively. 

Appendix A show the individualized patient analysis, which was possible as we had a longitudinal cohort. They depict the DSC variation for all OARs in each one of the eight patients in the independent test group with the semi-supervised model (Appendix A) and self-supervised model (Appendix A). For most OARs, the models performed slightly better for some patients than for others. On the contrary, the DSC of the larynx, lips, and submandibular glands showed large variations in performance depending on the CT and patient. For the submandibular glands, there was one patient for which the contours delineated by the radiation oncologists and the ones predicted by the model overlapped by less than 10% (Figure 4e–h). Appendix A also illustrates the improvement in DSC with the self-supervised model compared to the semi-supervised one (Appendix A).

## 4. Discussion

This paper proposes a DL-based workflow to automatically delineate fifteen OARs for HN radiotherapy treatment planning. We exploited a longitudinal partially labeled database to (i) generate accurate OAR segmentations, and (ii) demonstrate the possibility of enriching the model robustness with a baseline image from each patient.

In the pseudo-contouring step, we trained eleven OAR-specific models to generate pseudo-contours to complete the missing labels in the dataset (Table 1). As our data were longitudinal, including images from a patient in the training group was highly beneficial for robustly segmenting missing labels for that patient. The models showed high performance in the independent test for most OARs except the larynx, lips, and submandibular glands (DSC < 70% and ASD > 2 mm). Four OARs (spinal canal, esophagus, temporomandibular joints, and inner ears), showed a meaningful improvement in performance after data curation (Table 2), as the DSC increased by an average of 14.67% and the ASD decreased by a mean of 0.788 mm. Even if the models’ performances were not entirely satisfactory for some OARs, it must be considered that the goal of this workflow step was to complete the database by generating pseudo-contours with a simple training architecture. The reduced-data experiments demonstrated similar performance in the independent test between the original and reduced-data models (Figure 2). Therefore, we demonstrated that our approach for this first segmentation step was feasible in longitudinal databases with between 30 and 40 patients.

The generated pseudo-contours were then ensembled together with the available ground truth delineations, building a multi-class segmentation for each CT scan to train nnU-Net. After data refinement, the average DSC, sDSC, and ASD were 80.59%, 88.74%, and 0.904 mm, respectively (Table 3 (a)). These values supported the robustness of the generated pseudo-contours and the advantages of integrating spatial anatomical relationships during training. The OARs with the largest differences between the DSC and sDSC were the smaller structures: brainstem, parotid glands, temporomandibular joints, and inner ears. These OARs reached an average sDSC above 96%, indicating that almost all contour surfaces were within the specified tolerance and, therefore, would be exempted from corrections. These examples indicate the limitations of the commonly employed volumetric DSC [3,4] and support why we should start to asses OAR segmentation performance with other metrics such as sDSC, dosimetric calculations, time-based measurements, and clinical acceptability. The DSC does not assess the surface fraction that needs to be redrawn, as it does not account for whether there are numerous minor surface deviations across the volume or only a significant deviation at a single point, largely penalizing small volumes. Thus, comparing the results for large HN OARs, such as the brain of the spinal canal, with smaller OARs, such as the inner ears or the temporomandibular joints, results in unfair assessments. The sDSC was therefore of great aid to contrast the results of the DSC and get a broader evaluation. The larynx, lips, and submandibular glands obtained the worst performance overall, in agreement with other published papers (Table 4). 

These results showed that our model was comparable to previously published studies for HN OAR segmentation regarding the DSC [19] (Table 4), and also to other studies using partially labeled or unlabeled datasets and following similar approaches to ours [7,11,25]. Almost all OARs reached a performance close to the maximum DSC reported in the literature, outstanding in some cases such as the temporomandibular joints and inner ears. The lips were the only OAR that showed a clear underperformance. However, the only study that has evaluated this organ obtained a DSC of 71%, which demonstrates the difficulties in segmenting that region. 

Performance boost after data curation was evident in Figure 3, supporting the need for homogeneity in the segmentation protocol as also seen in previous works [30,31]. This problem was solved in most studies by requesting a single radiation oncologist to perform or review all the ground truth segmentations. Nonetheless, this is a highly time-consuming process, and not all medical centers have the resources to do it. As our objective was to use and exploit the available longitudinal partially labeled data in a hospital, we only reviewed and curated the contours that initially showed visual inconsistencies. Regardless, two other OARs revealed significant variations in performance during evaluation: larynx and lips. These OARs are especially complicated to segment (Table 4) and exceptionally in our cohort. On the one hand, many patients had massive tumors near the larynx, hindering its delineation. On the other, some patients were also intubated (Figure 4g), which modified lips shape and hampered their segmentation. Differences in these OARs’ ground truth segmentation protocol were also noted. 

We analyzed the results per patient (Appendix A), which was only possible as our data was longitudinal, allowing us to compare the contours in several CT scans for each patient. This demonstrated meaningful differences in performance between patients for particular OARs and even between CTs from the same patient. This was probably a consequence of the anatomical changes caused by the presence of large tumors or common secondary effects of RT treatment, which could be another source of underperformance in our DL models. For the submandibular glands, there was a patient for which the predicted and ground truth contours overlapped by less than 10% (Figure 4e–h). The question, in this case, was which one of the delineations is actually the more precise one. In the future, we aim to conduct a blinded test with an expert radiation oncologist from a different medical center to evaluate the ground truth and predicted contours for the patients in the independent test set and analyze which segmentations are more clinically acceptable.

Previous works [10,44] have shown how implementing DL-based OAR segmentation algorithms in new medical centers without retraining renders less agreement between predictions and ground truth segmentations. These discrepancies usually come from deviations in segmentation protocol [14], medical imaging machines with different technical characteristics [44], and overfitting to the training data. Therefore, it is evident that training DL-based models with data from the hospital where they will be implemented is very advantageous. To avoid the high computational cost of training a DL network from the beginning and the restricted sizes of homogenized labeled datasets, a more efficient approach would be performing transfer learning with pre-trained DL models. This reduces the computational requirements and includes additional data variability in the model. A further step, particularly valuable in adaptive radiation therapy, is to retrain the last epochs of a model with fully labeled CT scans from new patients being treated to teach the model to segment future CT scans from those patients with higher accuracy. Table 3b and Table 4 illustrate the significant performance improvements with this self-supervised approach, especially for the larynx, lips, and submandibular glands, which were the most difficult OARs to segment. Appendix A also depicts the DSC improvement when using the self-supervised model (Appendix A), demonstrating the capability of the self-supervised model to yield more robust and personalized predictions.

The proposed DL-based workflow can automatically contour fifteen OARs in the clinic. It has proven to be an efficient way to exploit the available partially labeled data from a medical center and presented a more realistic solution than studies that rely on large databases segmented from the beginning. Our model showed comparable performance to the literature, with a more straightforward and pragmatic implementation in new scenarios. Our solution was developed using CT scans derived from routine clinical practice, belonging to patients with advanced local tumors, and therefore should be applicable in a hospital setting. Moreover, we demonstrated the feasibility of integrating our workflow for adaptive RT, yielding robust patient-adapted contours requiring fewer human and computational resources. 

However, there were some limitations to this study. First, the cohort included only 48 unique patients. Although we applied numerous data augmentation tools to increase data variability, only 40 unique patients were used for training. Moreover, the contours of the larynx and lips were not refined to ensure homogenization in the segmentation protocol, possibly downgrading model performance. Another limitation is that we reused the tolerances to compute the sDSC described by Nikolov et al. [3] instead of calculating our own values.

For future studies, we would like to extend the proposed workflow to segment more HN OARs, such as the oral cavity, constrictor muscles of the pharynx, or optic nerves. We believe that a mandatory step prior to training must be to curate the data under a homogeneous protocol and evaluate if there are any changes in already segmented OARs such as the lips or larynx. Furthermore, we would like to conduct a blinded test with an expert radiation oncologist from a different medical center to analyze which segmentations are clinically acceptable, comparing both ground truth and predicted contours.

## 5. Conclusions

This paper presents a novel DL-based weakly supervised workflow for HN OAR segmentation exploiting partially labeled datasets and longitudinal data. Experimental results showed its superior performance for fast segmentation of fifteen OARs of the head and neck anatomy. The model can be easily integrated into the clinical environment as it has been trained with data derived from clinical practice. Our results also show the feasibility of implementing the workflow for adaptive radiotherapy, contributing to the availability of DL-based auto-segmentation tools for clinical users in the near future.

## Figures and Tables

**Figure 1 entropy-24-01661-f001:**
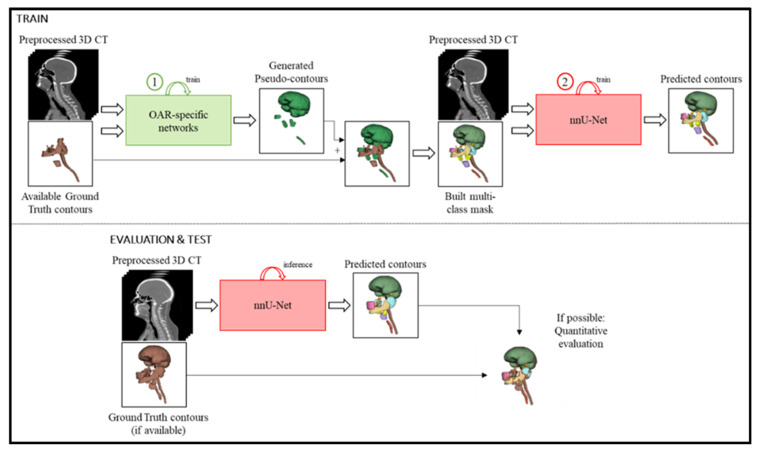
Proposed framework. During training, the CT image is first preprocessed and curated. Then, the OAR-specific networks are trained to generate the pseudo-contours, which are ensembled together with the available ground truth segmentations into a multi-class mask. These fully segmented images are used to train nnU-Net to predict the OAR contours. During inference, it is only necessary to preprocess the CT image and predict contours with the trained nnU-Net. The results can then be evaluated and compared to the ground-truth delineations (if available). (Ground truth contours make up a multi-class mask but are depicted in brown for visualization purposes).

**Figure 2 entropy-24-01661-f002:**
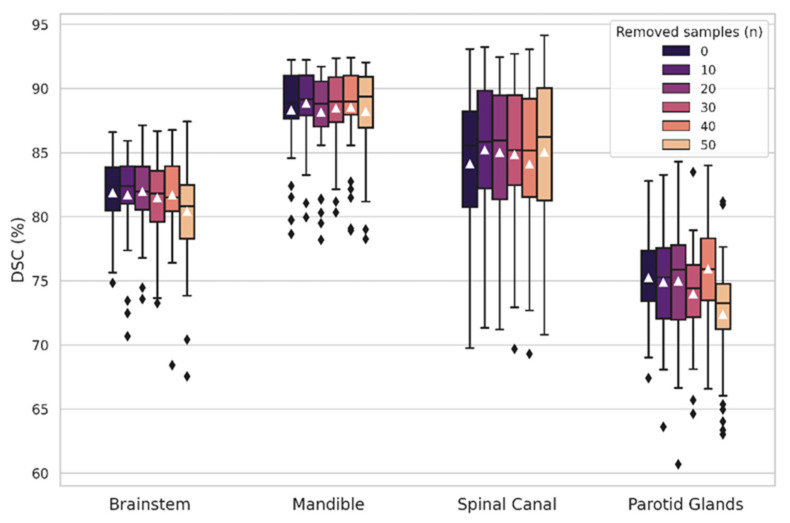
Box and whiskers plots of the Dice score coefficient (DSC) between the ground truth and predicted labels for the brainstem, mandible, spinal canal, and parotid glands with the original OAR-specific and reduced-data models. The first boxes correspond to the original models, and the consecutive ones depict the resulting DSC after removing 10, 20, 30, 40, and 50 training samples.

**Figure 3 entropy-24-01661-f003:**
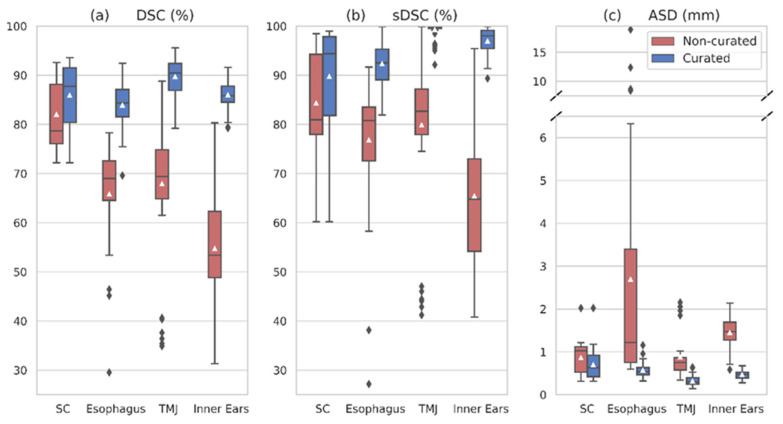
Box and whiskers plots of the performance metrics between the predicted and ground truth contours: (**a**) Dice score coefficient (DSC); (**b**) surface Dice similarity coefficient (sDSC); (**c**) average surface distance (ASD). In each plot, the comparison between the results with the uncurated (red) and curated (blue) data can be observed for the four OARs that underwent clinical curation. (SC: spinal canal; TMJ: temporomandibular joints).

**Figure 4 entropy-24-01661-f004:**
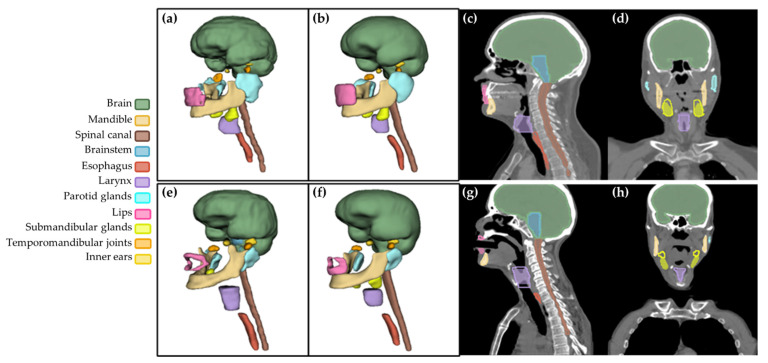
The first row corresponds to the best-segmented patient (Test 01) in the independent test set: (**a**) 3D ground truth (GT) contours, (**b**) 3D predicted contours, and comparison between them in (**c**) the sagittal and (**d**) coronal planes. The second row shows the worst segmented patient (Test 03): (**e**) 3D GT contours, (**f**) 3D predicted contours, and comparison between them in the (**g**) sagittal and (**h**) coronal planes. For this patient, the DSC was lower than 10% for the submandibular glands between the GT and model’s predictions. In (**c**,**d**,**g**,**h**), the shaded segmentations correspond to the predictions, and the outlined borders to the GT delineations. The visualization was performed in 3DSlicer 4.10.2.

**Table 1 entropy-24-01661-t001:** OAR contouring frequency in the dataset used to train the workflow. # stands for “Number of”. The first column depicts the OARs segmented in the study. The second and third columns correspond to the total number of patients for which each OAR segmentation was available in at least one CT, and the number of CTs contoured for that OAR, respectively. These CTs were then divided into train/test sets for the OAR-specific models. The division is depicted in the following two columns.

OAR	# Annotations Used to Train the Workflow		# CT Images in OAR-Specific Models	
	Patients (n = 40)	CTs (n = 225)	n_train_	n_test_
Brain	33	165	132	33
Mandible	40	224	179	45
Spinal canal	40	221	176	45
Brainstem	40	225	180	45
Esophagus	34	179	143	36
Larynx	36	198	158	40
Parotid glands	40	217	173	44
Lips	33	181	144	37
Submandibular glands	34	183	146	37
Temporomandibular joints	38	205	164	41
Inner ears	39	217	173	44

The data in the independent set, comprising 44 CT images from 8 patients with all OARs segmented, was kept aside from all training steps.

**Table 2 entropy-24-01661-t002:** Dice score coefficient (DSC) and average surface distance (ASD) from evaluating the OAR-specific models in each individual test set, uncurated and curated independent test group. The number of CT images (n_CT_) used for training and testing each network is also depicted. (PG: parotid glands; SMG: submandibular glands; TMJ: temporomandibular joints).

OAR	Test		Independent Test (n_CT_ = 44)			
	DSC (%)	ASD (mm)	DSC (%) Uncurated	DSC (%) Curated	ASD (mm) Uncurated	ASD (mm) Curated
Brain	98.04 ± 1.19	0.440 ± 0.070	98.16 ± 0.29		0.494 ± 0.088	
Mandible	91.09 ± 1.75	0.411 ± 0.078	88.31 ± 3.42		0.553 ± 0.132	
Spinal canal	88.87 ± 4.54	0.903 ± 1.302	80.21 ± 6.53	84.11 ± 5.67	1.007 ± 0.454	0.833 ± 0.401
Brainstem	89.07 ± 4.19	0.636 ± 0.225	81.86 ± 2.76		1.084 ± 0.214	
Esophagus	81.39 ± 4.61	0.844 ± 0.492	63.07 ± 11.87	73.50 ± 10.36	2.818 ± 3.178	1.116 ± 0.743
Larynx	82.64 ± 9.11	0.990 ± 0.545	66.73 ± 9.31		2.026 ± 0.753	
PG	86.05 ± 5.79	1.162 ± 2.701	75.22 ± 3.55		1.113 ± 0.246	
Lips	71.88 ± 15.11	2.177 ± 3.801	45.17 ± 18.18		2.346 ± 1.748	
SMG	79.39 ± 7.72	0.782 ± 0.331	55.40 ± 21.14		2.489 ± 2.475	
TMJ	87.31 ± 3.57	0.430 ± 0.201	67.26 ± 12.36	83.34 ± 4.72	0.916 ± 0.467	0.523 ± 0.132
Inner ears	87.39 ± 3.03	0.514 ± 0.703	55.49 ± 12.98	83.76 ± 3.69	1.414 ± 0.398	0.531 ± 0.129

**Table 3 entropy-24-01661-t003:** Dice score coefficient (DSC), surface Dice similarity coefficient (sDSC), and average surface distance (ASD) corresponding to the evaluation of trained nnU-Net in the independent test with the (a) semi-supervised model; (b) self-supervised model. For the four refined OARs, the results are shown for the uncurated data (U) and curated one (C). (PG: parotid glands; SMG: submandibular glands; TMJ: temporomandibular joints).

OAR		(a) Semi-Supervised Model			(b) Self-Supervised Model		
		DSC (%)	sDSC (%)	ASD (mm)	DSC (%)	sDSC (%)	ASD (mm)
Brain		97.99 ± 0.29	96.68 ± 1.27	0.475 ± 0.067	98.02 ± 0.29	96.97 ± 1.17	0.470 ± 0.068
Mandible		90.10 ± 2.76	96.46 ± 1.72	0.465 ± 0.108	90.60 ± 2.34	97.21 ± 1.38	0.440 ± 0.082
Spinal canal	U	82.03 ± 6.74	84.33 ± 10.61	0.864 ± 0.364	83.53 ± 6.03	87.26 ± 8.58	0.822 ± 0.354
	C	85.95 ± 6.25	89.75 ± 10.29	0.696 ± 0.339	86.92 ± 5.04	91.86 ± 7.29	0.669 ± 0.287
Brainstem		86.14 ± 3.32	98.15 ± 4.27	0.827 ± 0.184	87.05 ± 3.63	98.33 ± 4.71	0.776 ± 0.194
Esophagus	U	65.83 ± 12.19	76.86 ± 12.25	2.694 ± 3.552	66.04 ± 13.92	77.11 ± 14.15	2.987 ± 3.929
	C	83.92 ± 4.56	92.35 ± 4.27	0.569 ± 0.156	86.77 ± 4.43	95.24 ± 3.53	0.507 ± 0.139
Larynx		71.57 ± 7.00	73.73 ± 12.15	1.630 ± 0.779	79.47 ± 6.50	85.67 ± 11.16	1.254 ± 0.719
PG		82.56 ± 2.81	96.29 ± 2.37	0.802 ± 0.142	83.70 ± 2.20	97.48 ± 1.84	0.753 ± 0.118
Lips		51.20 ± 15.80	58.05 ± 18.76	1.449 ± 0.420	68.96 ± 9.09	78.83 ± 13.28	0.957 ± 0.301
SMG		61.29 ± 25.61	78.44 ± 26.64	2.245 ± 2.690	65.80 ± 22.34	83.79 ± 22.04	1.912 ± 2.223
TMJ	U	67.92 ± 14.16	79.89 ± 15.96	0.877 ± 0.489	67.83 ± 14.92	79.16 ± 16.18	0.882 ± 0.546
	C	89.73 ± 3.81	99.27 ± 1.62	0.321 ± 0.119	91.38 ± 2.17	99.59 ± 1.02	0.256 ± 0.073
Inner ears	U	54.74 ± 12.69	65.43 ± 14.35	1.446 ± 0.378	53.12 ± 13.42	61.71 ± 16.18	1.534 ± 0.416
	C	86.03 ± 2.88	96.98 ± 2.77	0.467 ± 0.097	88.16 ± 2.56	98.18 ± 2.03	0.414 ± 0.094

**Table 4 entropy-24-01661-t004:** Mean volumetric Dice score coefficient (DSC) performance of previously published deep-learning models for HN segmentation on CT and our model. Due to the large volume of publications, this overview includes only meaningful results. The datasets and ground truth segmentations used vary between studies making comparison difficult. We include the results for our self-supervised model in the last row of the table. (PG: parotid glands; SMG: submandibular glands; TMJ: temporomandibular joints).

	Brain	Mandible	Spinal Canal	Brainstem	Esophagus	Larynx	PG		Lips	SMG		ATM		Inner Ears	
							L	R		L	R	L	R	L	R
Nikolov (2018) [3]	99	96	95	88			85	85		85	85			65	75
Zhu (2019) [32]		93		87			88	87		81	81				
Liang (2019) [23]		91		90		87	85	85				85	84		
van Rooij (2019) [13]									64	60	78	83	83		82
Tang * (2019) [25]		93							86		89	85	85		81
Zhong (2019) [22]							92								
van Dijk (2020) [15]		95 ^1^		83 ^1^			84 ^1^	83 ^1^		77 ^1^	78 ^1^				
Guo (2020) [27]		95		88			88	88		84	84				
Brunenberg (2020) [44]		90		78			83	83		79	78				
Sultana (2020) [24]							87	86		87	85				
Oktay (2020) [10]		94		85			84	85		83	78				
Chi * (2020) [7]		88					73	73		63	61				
Zhang (2021) [26]		89		87			71	77				70	70		
Liu * (2021) [11]	97	92	89	86		90	84	87				90	90		
Dai (2021) [45]		89		90	85	88	83	82						67	67
Zhang (2021) [46]		95		92			86	88		87	85				
Li (2022) [12]						84									
Tappeiner (2022) [47]		94		88			88	87							
Siciarz (2022) [48]	97	89	86	87	84	86	80	81	71	77	76			72	74
Koo (2022) [21]		87		88	82	82	83	83		81	83				
Gibbons (2022) [49]		91 ^2^		83 ^2^	48 ^2^		80 ^2^	80 ^2^		69 ^2^	67 ^2^				
Asbach (2022) [50]	93			85		43	81	81		72	70			45	47
OURS	98	90	86	86	84	72	83		51	61		90		86	
OURS—Self-Supervised	98	91	87	87	87	77	84		69	66		91		88	

^1^ Values estimated from figures; actual values not reported. ^2^ Values correspond to median DSC. Mean DSC not reported. * Studies using partially labeled and unlabeled datasets.

## Data Availability

The datasets analyzed during the current study are not publicly available due to the restrictions imposed by the ARTIX study.

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
