# Peer review of "Deep Learning-Based Segmentation of Head and Neck Organs-at-Risk with Clinical Partially Labeled Data"

_entropy, 2022, doi:10.3390/e24111661_

Round 1

Reviewer 1 Report

The research aims at solving a fundamental problem of organ delineation that could prove handy for doctors and everyone else.

The literature review of the paper can be improved and should be moved to a separate section titled a literature review.

The authors haven’t provided many details about the research flow.

The authors should consider drawing a research diagram to explain their research flow.

Future works should be listed to the paper.

Metrics like Dice Score Coefficient (DSC), Surface Dice Similarity Coefficient (sDSC), and Average Sur- 317 face Distance (ASD)should  be defined for readers.

Authors need to highlight any changes they made to nnU-Net(if any)

Reviewer 2 Report

General comments:

Well-written. However, make sure that you do not change too much between past and present tense (I commented a bit on this in the beginning of the text, but stopped again, please correct yourselves).

More detail and emphasis are needed in section 2.2 to point out that it is a two-step process. This description is way too short.

Specific comments:

L11-12: “Manual delineation is time, labor-consuming, and observer-dependent.” -> Either “Manual delineation is time-consuming, labor-consuming, and observer-dependent” or “Manual delineation is time- and labor-consuming, as well as observer-dependent.”

L13: Potentially “large and homogeneously contoured databases” -> “large databases of homogeneously contoured image sets”

L18-19: “longitudinal OAR-specific 3D models for pseudo-contour generation followed by a simultaneous OARs segmentation based on a multi-class nnU-Net” This sentence contains too many non-standard terms.

L20: “the superior performance” -> “superior performance” or “a superior performance”

L21: “DSC” and “sDSC” these abbreviations need to be introduced

L21-22: Potentially “80.59%” and “88.74%” -> “81%” and “89%”. In general (i.e. in the rest of the text), it might be too much with two decimals for DSC values

L22: “We demonstrate” -> “We demonstrated”

Abstract: How many patients were included in the test and evaluation set?

L33: Potentially “prescriptions” -> “treatments” or “treatment strategies”

L37: “(organs at risk)” -> “(organs at risk (OAR))”

L39-42: “This allows estimating the dose these structures will receive to develop personalized strategies to mitigate the radio-induced toxicities [3], critical in HN cancer due to the high amount of OARs present in the region .” this sentence is somewhat incomplete. Potentially: “This allows estimating the dose these structures will receive and to develop personalized strategies to mitigate the radio-induced toxicities [3], which is critical in HN cancer due to the high amount of OARs present in the region.”

L49-51: “The poor image quality of computed tomography 49 (CT) images is also a substantial problem in HN delineation [10], [11], as many OARs have similar densities to fat, muscle, or other surrounding tissues.” This is not an image quality issue, but rather a soft tissue contrast issue (at least I don’t think it is fair to state that CT image quality is general poor)

L60: Potentially “in the segmentations” -> “in the automatic segmentations”

L75: Potentially “an extensive database” -> “an extensive database of patient images”

L85: “propose a” -> “proposed a”

L86-87: “a self-supervised 2D CNN to segment CT 2D projections with a contrastive loss.” 1) Is it really CT projection data which is segmented and not rather the 2D CT image slices? 2) I guess “a contrastive loss” is a description of the CNN rather than of the CT projections (/ images), and if this is correct it should be “a self-supervised 2D CNN with a contrastive loss function to segment CT 2D projections / 2D CT image slices”

L87: “200 CTs” Is this 200 CT images (i.e. 200 3D image stacks) or 200 CT image slices? (Also, how many patients does it refer to? 200 unique patients?)

L90: Introduce abbreviation “DSC”

L96: “OARs training” -> “OARs by training” or something similar

L99: “updating of different” should this have been “updating based on different”?

L85-100: See my general comment above about being consistent in the past/present tense. In general, past tense should be used, and particular in a paragraph like this were you are describing previous work by others.

Introduction: The Introduction section is very long and very detailed (I would suggest to shorten it), but I don’t know the journal guidelines

L115-116: Potentially “Each patient’s contours were delineated by one expert radiation oncologist amongst a team of ten who participated in the whole study.” -> “For each patient all contours were delineated by a single expert radiation oncologist.”

Table 1: The information of the independent test set is weirdly placed in the table.

L133: “randomly selected” this is not random when you had specific criteria, therefore delete “random”

L141: “nnU-Net” more info needed, or a reference to a description of what a nnU-Net is

L137-142: This is very important info, but it is very short, so it is hard to follow that this is actually a two-step process (and this makes it hard to understand the rest of the manuscript).

Figure 1: Also in this figure the two-step process needs to be emphasized more clearly.

Figure 1: This figure does not seem to be referenced in the text.

Figure 1: There is some white text on the CT image you use. Remove this and the white/grey stripes in the top/right part of the image. The blue contour overlay in the left images is difficult to understand, and for the test set I guess it should not be there, since these images should not be delineated before applying the network

L164: Potentially “the same mask” -> “one mask”

L165: “Several ablation studies performed by our group” 1) what is meant by “ablation” (it can mean to surgically remove tissue with high and focused energy, but I don’t assume that this is what you mean here)? 2) If you here refer to previously published results, then you should bring a reference, and if this is not published you should describe it in more detail potentially in a supplementary, since right now it is just a postulate.

L168-174: You could potentially sketch the network in a figure in the supplementary unless you can cite another paper where it has already been visualized

L179: “MONAI” potentially add a few more words to explain what this is (is it a software? Or?)

L180 “each epoch” -> “each epoch of the training”

L189: “Loss function” -> “The loss function”

L194: “evaluated in their own test set and in the independent test group” 1) do “their own test set” and “the independent test group” refer to two different things? If yes, I think some info about this is lacking (or have I overlooked this above?). 2) Potentially “evaluated on” and “and on”

L195: “Average Surface Distance (ASD)” I don’t know this quantity, maybe good to bring a reference which explains how this is computed

L200: “which fewer ground truth” -> “which the fewest ground truth”

L207-208: Potentially “pseudo-contours to replace the missing labels in the cohort” -> “pseudo-contours for the CT images with missing labels/contours” and potentially add “to have a fully segmented training image set”

L212: 8 seconds per contour or for all contours or for all missing contours (also, did you create pseudo-contours when a ground-truth contour already existed?)

L214: “eleven classes” above you mentioned 15 OARs. Is this because some of the left/right structures were put into one? (If yes, potentially good to mention above that you ended up with 11)

L219-220: “As input we used the ensembled multi-class contours” what about the CT scans, were they not used?

L221-222: Potentially “anatomical spatial relationships” -> “anatomical spatial relationships between the individual OARs”

L225: “23 hours” potentially add the full training time

Table 2: When only one value is present for “uncurated” and “curated” is it then because the values are the same for the two, or because only “uncurated” data is available? In the latter case, probably most fair to just state the result under “uncurated” and leave “curated” blank

L260-261: “After extracting 50 images, a slight decrease in the DSC could be observed for 260 the brainstem and the parotid glands” potentially mention how many images were left in the training data after removing 50 images

L271: “external test group” you haven’t used the term “external” before. Is this test group stemming from a different hospital?

L273: “after refinement” potentially “after curation of the ground truth contours”

L287-288: “The sDSC was higher than the DSC for all OARs except the brain, for which the average sDSC and DSC were 96.66% and 97.99%, respectively.” Why is this important? Or rather, which conclusion can be drawn based on this?

Supplementary material: To ease the reading of the supplementary, you ought to collect the two figures in one PDF document together with their captions (and potential other material suggested in this review)

Figure 4: Several colors are very similar and cannot be distinguished

L350: “smaller than” -> “lower than”

L351-352: “the background segmentations correspond to the predictions, and the outer contour to the GT delineations” What is meant by “background segmentations” and “outer contour”? Do you mean the shaded and the border, respectively?

L403: Potentially “alongside previous works” -> “as also seen in previous works”

Discussion section is very long and repeats a lot of information from the MM and Results section
